# New Caledonian crows keep 'valuable' hooked tools safer than basic non-hooked tools

Barbara C Klump[1,2]*, James JH St Clair[1], Christian Rutz[1]*

[1]Centre for Biological Diversity, School of Biology, St Andrews, United Kingdom; [2]Cognitive and Cultural Ecology Group, Max Planck Institute of Animal Behavior, Radolfzell am Bodensee, Germany

**Abstract** The temporary storage and re-use of tools can significantly enhance foraging efficiency. New Caledonian crows in one of our study populations use two types of stick tools – hooked and non-hooked – which differ in raw material, manufacture costs, and foraging performance. Using a large sample of wild-caught, temporarily captive New Caledonian crows, we investigated experimentally whether individuals prefer one tool type over the other when given a choice and whether they take better care of their preferred tools between successive episodes of use, safely storing them underfoot or in nearby holes. Crows strongly preferred hooked stick tools made from *Desmanthus virgatus* stems over non-hooked stick tools. Importantly, this preference was also reflected in subsequent tool-handling behaviour, with subjects keeping hooked stick tools safe more often than non-hooked stick tools sourced from leaf litter. These results suggest that crows 'value' hooked stick tools, which are both costlier to procure and more efficient to use, more than non-hooked stick tools. Results from a series of control treatments suggested that crows altered their tool 'safekeeping' behaviour in response to a combination of factors, including tool type and raw material. To our knowledge, our study is the first to use safekeeping behaviour as a proxy for assessing how non-human animals value different tool types, establishing a novel paradigm for productive cross-taxonomic comparisons.

## Editor's evaluation

The authors show experimentally that New Caledonian crows, a rare tool-using bird, prefer hooked stick tools for foraging and safely store them underfoot or in holes. Quantifying safekeeping behaviour helps understand how non-human, tool-using animals value different tool types.

## Introduction

Humans have become completely dependent on the use of tools. Every day, we use a multitude of objects for interacting with our environment. While we consider many tools disposable, we greatly value others and look after them when they are not needed: a treasured pen is carefully placed in a pen holder, a trusted hammer is secured to a tool belt, and an expensive electronic device is stored in a padded pouch. Such 'safekeeping' of tools has at least two main functions. First, it ensures that we can easily find these tools when we need them, avoiding search costs, and second, it minimises the likelihood of tool damage or loss, avoiding replacement costs. To our knowledge, there are no studies that investigated if non-human animals also handle 'valuable' tools more carefully.

Using tools for foraging has the benefit of providing access to some nutritious food sources that would otherwise be difficult or impossible to obtain, such as embedded arthropods, honey in tree

*For correspondence:
bklump@ab.mpg.de (BCK);
christian.rutz@st-andrews.ac.uk (CR)

cavities, or the content of hard-shelled items like nuts and eggs (*Bentley-Condit and Smith, 2010*; *Shumaker et al., 2011*; *Sanz et al., 2013*). But tool use also requires investment in terms of time and energy. A suitable tool needs to be found, or raw materials sourced for tool manufacture, which can be time-consuming if the preferred items are scarce in the environment (*Koops et al., 2015*; *Almeida-Warren et al., 2017*). Tool manufacture and/or modification can also be demanding, requiring extensive processing of materials (*Hunt and Gray, 2004*; *Klump et al., 2015a*; *Lapuente et al., 2017*), and if mistakes are made, procurement of fresh raw materials (*Tebbich et al., 2012*). And finally, during tool deployment, the animal may be at risk of losing its tool, either to thieving conspecifics or by accidentally dropping it (*Nishida and Hiraiwa, 1982*; *Tebbich et al., 2012*; *Klump et al., 2015b*). Animals can offset some of these costs by re-using tools and by keeping them safe between periods of use, either holding onto or storing them. Such tool 'safekeeping' has indeed been observed anecdotally in otters (*Hall and Schaller, 1964*), chimpanzees (*Nishida and Hiraiwa, 1982*), and Galapagos woodpecker finches (*Tebbich et al., 2012*), and first controlled studies have examined the behaviour's context-dependent expression in New Caledonian crows *Corvus moneduloides* (hereafter 'NC' crows) (*Klump et al., 2015b*) and Goffin's cockatoos (*Auersperg et al., 2017*). Specifically, we have previously shown experimentally that NC crows respond to an increase in tool recovery costs (foraging at height) with elevated levels of safekeeping behaviour (*Klump et al., 2015b*), a result that was subsequently replicated in Goffin's cockatoos (*Auersperg et al., 2017*) – a species that does not seem to routinely use tools in the wild (*O'Hara et al., 2021*). Interestingly, when NC crows' preferred safekeeping method of holding tools underfoot was made more challenging, because subjects had to handle demanding prey before re-using their tools, they resorted to storing tools more frequently in holes, thereby preventing their accidental loss (*Klump et al., 2015b*). It remains unknown, however, whether NC crows' safekeeping behaviour is also sensitive to a tool's manufacture costs and/or its potential utility in terms of foraging efficiency. In other words, do NC crows keep 'valuable' tools safer than more basic ones?

NC crows in one of our study populations ('farmland' site in *Rutz et al., 2012*) use two different types of stick tools. Non-hooked stick tools (*Figure 1a*, top panel) are twigs or leaf petioles sourced from the forest floor, or branches removed from live vegetation (*Hunt and Gray, 2002*); they are ubiquitous, abundant, and usually require no modification before they can be used (*Rutz and St Clair, 2012*). Hooked stick tools (*Figure 1a*, bottom panel), on the other hand, are almost always made from one particular plant species, the shrub *Desmanthus virgatus*, and birds need to locate this patchily distributed material, select a suitable stem, and manufacture the tool in an elaborate process, including the careful 'sculpting' of a hooked tip (*Hunt and Gray, 2004*; *Klump et al., 2015a*; *St Clair et al., 2016*; *Klump et al., 2019*). Although tool procurement costs have not yet been quantified in NC crows, it is evident that hooked stick tools will on average require a greater investment of time in both search and manufacture behaviours than non-hooked stick tools (for further discussion, see 'Concluding remarks'). In an earlier study, we compared the foraging performance of crows using different tool types on standardised food extraction tasks and found that self-manufactured hooked stick tools made from *D. virgatus* were much more efficient than non-hooked stick tools sourced from leaf litter (*St Clair et al., 2018*). Given the extra costs involved in sourcing suitable material and manufacturing a hooked stick tool (plant material requires elaborate processing [*Klump et al., 2015a*]; producing more efficient, deep-hooked tools often involves additional actions [*Sugasawa et al., 2017*]; choosing the wrong plant species may be costly [*Klump et al., 2019*]) as well as the enhanced foraging performance of this tool type (*St Clair et al., 2018*), we predicted that NC crows would prefer these tools over non-hooked ones when given a choice, and 'value' them more highly, taking better care of them when not in active use. In the present study, we tested these predictions in two companion experiments with wild-caught, temporarily captive crows.

In Experiment 1, we offered subjects choices of hooked and non-hooked stick tools, and recorded which tool type they selected for a single prey extraction. Experiment 2 examined whether crows treated tool types differently between successive prey extractions, and also (in separate treatments) manipulated tool material and manufacture costs. The first two treatments in Experiment 2 (2.A and 2.B, see *Figure 1b*) were designed to mimic natural foraging conditions, providing a comparison between the safekeeping of non-hooked stick tools crow-sourced from assorted twigs and leaf petioles, and hooked stick tools crow-manufactured from *D. virgatus*. While this achieved good ecological validity, it inevitably confounded tool type (non-hooked vs. hooked) with material (twigs and

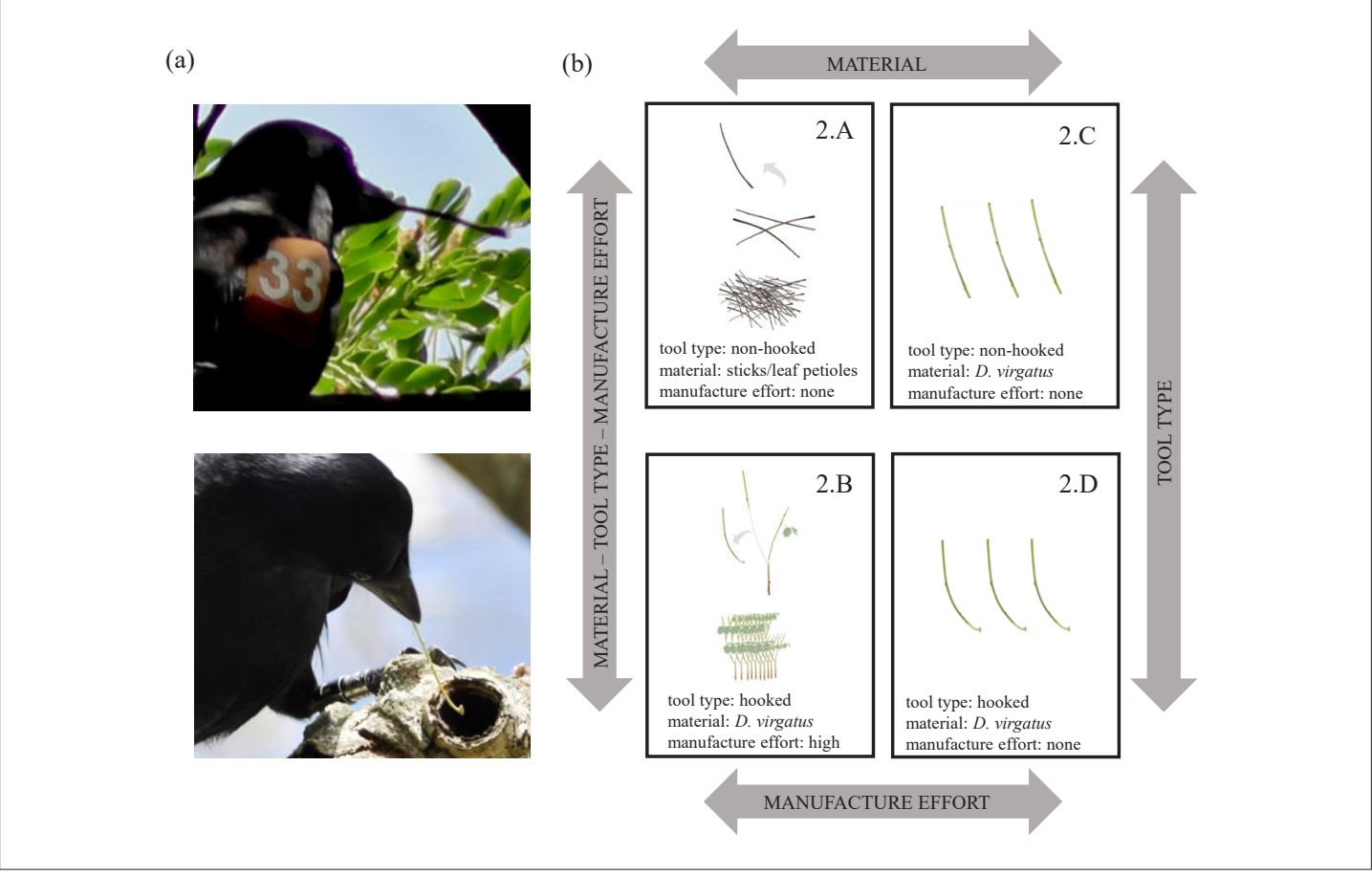

**Figure 1.** Tool types used by New Caledonian crows and experimental treatments of Experiment 2. (**a**) Crows in our study population are known to use two tool types – non-hooked stick tools (top) and hooked stick tools (bottom). (**b**) Treatments in Experiment 2. To assess crows' tool-placement behaviour following successful food extractions, we presented them with different materials: 100 sticks and leaf petioles (Treatment 2.A), 10 stems of *Desmanthus virgatus* (the preferred raw material for hooked stick tool manufacture in this population) with multiple forks suitable for tool manufacture (Treatment 2.B), 3 non-hooked stick tools researcher-made from *D. virgatus* (Treatment 2.C), and 3 hooked stick tools researcher-made from *D. virgatus* (Treatment 2.D). Arrows indicate differences between treatments, and the layout of treatments is the same as in **Figure 3**, to facilitate comparison. Figure adapted from **St Clair et al., 2018**, but note different labelling of treatments (2.A in the present study → 1b in earlier study; 2.B → 1a; 2.C → 2b; 2.D → 2a).

leaf petioles vs. *D. virgatus*) and manufacture effort (no manufacture vs. elaborate processing of raw material). We therefore ran two additional treatments with researcher-supplied tools (2.C and 2.D, see *Figure 1b*) to test whether any of these aspects by themselves affect crows' tool-placement behaviour.

To our knowledge, our study is the first to use animals' tool-handling behaviour to make inferences about the value they ascribe to different tool types. This novel experimental paradigm allows an assessment of how animals value tools without the need for elaborate training, unlocking considerable research potential for observational and experimental studies across a wide range of tool-using species (including those for which safekeeping has already been reported anecdotally; see above).

## Materials and methods
### Study site and subjects
Between 17 September and 28 November 2012, 24 August and 28 October 2013, and 5 August and 8 October 2015, a total of 64 NC crows (12 of them in more than 1 year) were trapped non-selectively with meat-baited whoosh nets in a farmland area near Bourail, on the central west coast

of New Caledonia (for housing conditions and husbandry protocols, see *St Clair and Rutz, 2013*). Nine birds were released immediately as they were breeders or appeared to be in poor health, two birds escaped, and seven birds were released before experimental protocols were finalised (see below). In pre-testing sessions (for details, see *St Clair and Rutz, 2013*), 35 birds manufactured and used hooked stick tools and therefore progressed to the main experiments reported here (while hooked stick tool making seems geographically restricted, the use of non-hooked stick tools is widespread; *Hunt and Gray, 2002*). Of these, 8 failed to interact with the task in their first trial, leaving 27 birds that participated in either one or both of our experiments (see *Supplementary file 1a*). Twenty-three birds participated in Experiment 1 (5 in 2012, 9 in 2013, 10 in 2015; 1 adult female [CEO] was tested both in 2012 and 2013), and 17 birds participated in Experiment 2 (8 in 2012, 9 in 2013).

Following recommendations of the STRANGE framework for animal behaviour research (*Webster and Rutz, 2020*), we provide demographic details of the sample of subjects that contributed data to our analyses, as well as of the sample of birds that were excluded for the above reasons (see *Supplementary file 1a*; morphometric data do not seem relevant to the present analyses, and are therefore not reported, and we do not have information on subjects' social status, personality type, and experience). Based on these data, we conclude that neither the sample of trapped birds (binomial test: p=0.26, n = 64) nor the sample of subjects that contributed data to analyses for Experiment 1 and/ or Experiment 2 (binomial test: p=0.08, n = 26) was significantly sex biased, and that the sex and age composition of birds that contributed data did not differ significantly from that of birds that did not contribute data (Fisher's exact test for birds that contributed data vs. birds that did not contribute data; sex: p=0.20; age: p=0.56; see *Supplementary file 1b*). In order to check whether birds that contributed data to both experiments (see below) might have been a non-random sample of those that only contributed data to Experiment 1, we also compared how often birds in these groups chose the single hooked stick tool in Experiment 1 (for details, see below) and found no significant difference (eight Fisher's exact tests: pick-up, transport, deployment, and extraction for Treatments 1.A and 1.B, respectively; all p≥0.67).

All subjects were tested individually in an experimental aviary (for details, see *St Clair and Rutz, 2013*). To ensure motivation, food was removed from the housing aviary either the evening (for morning sessions) or ca. 1.5 hr (for afternoon sessions) before testing. During experimental trials, birds had *ad libitum* access to water, but no food except for the bait provided in the extraction tasks.

## Experimental set-up and procedures

### Experiment 1

Experiment 1 consisted of two treatments (Treatments 1.A and 1.B) that investigated whether our subjects prefer hooked over non-hooked stick tools (for an example of supplied tools, see *Figure 2—figure supplement 1*). In each treatment, 21 tools were presented on a circular platform: 20 non-hooked stick tools and 1 hooked stick tool (the non-hooked stick tools differed between treatments, as explained below). The majority of tools were non-hooked to reduce the probability that crows would pick the hooked stick tool simply by chance. All tools were within the length and diameter range reported for wild crow tools (*Bluff et al., 2010b*; *Sugasawa et al., 2017*). Tools within each treatment were matched in length (±1 cm), and all tools made from *D. virgatus* were additionally matched for stem colour.

In Treatment 1.A, we presented 1 hooked stick tool researcher-made from *D. virgatus*, 1 non-hooked stick tool researcher-made from *D. virgatus*, and 19 (non-hooked) sticks and leaf petioles sourced from leaf litter; in Treatment 1.B, we presented 1 hooked stick tool researcher-made from *D. virgatus*, 1 (non-hooked) leaf petiole, and 19 non-hooked stick tools researcher-made from *D. virgatus* (*Figure 2—figure supplement 1*). In each treatment, the inclusion of an 'anomalous' non-hooked stick tool, which differed from the remaining 19 tools, allowed us to control for the possibility that crows might simply select odd-looking tools, rather than paying attention to the presence of hooks. In each treatment, the positions of the hooked stick tool and the anomalous non-hooked stick tool on the platform were randomised, and the remaining tools were scattered loosely on top of these two tools, ensuring that all tools remained visible. Treatment 1.A was run in 2013 and 2015, and Treatment 1.B was run in 2012 and 2013. In 2013, both treatments were run back-to-back in the same session, in randomised order.

To incentivise crows to choose and use tools, a food log was provided with a single drilled hole (16 mm wide and 70 mm deep) baited with a peanut-sized piece of meat (henceforth referred to as 'prey'); a tool (either hooked or non-hooked) was required to extract the prey from the hole. A tiny piece of meat ('teaser') was positioned on the food log to attract the subject's attention, unless two trials were run back-to-back (in which case a teaser was only presented in the first trial; n = 3). Due to experimenter error, no teaser was presented in one trial, and in another one, we could not confirm from the video recording that a teaser had been presented. Trials lasted until the prey was extracted or for ca. 10 min, whichever occurred first, and were filmed with a Panasonic HD camcorder from a hide adjacent to the experimental aviary. At the end of each treatment, the observer called an assistant via radio. If a second treatment was run, the assistant removed all tools, placed the new tools on the circular platform out of view of the subject, and rebaited the food log in full view; otherwise, the session was terminated.

### Experiment 2

Data for this experiment originated from an earlier study comparing the foraging efficiency of different tool types (*St Clair et al., 2018*). Each subject participated in two key treatments, run on different days: one where the bird was provided with 100 non-hooked sticks and leaf petioles of assorted length scattered on a material log (Treatment 2.A), and a second, in which 10 *D. virgatus* stems with several forked branches (judged to be suitable for hooked stick tool manufacture) were presented wedged upright into the material log (Treatment 2.B). These two treatments approximate how crows procure tool materials in nature, but they inevitably confound tool type, material, and manufacture effort (see 'Introduction'). To disentangle the relative contributions of these factors, 8 of the subjects (all in 2012) participated in two additional treatments where three researcher-supplied tools made from *D. virgatus*, either non-hooked (Treatment 2.C) or hooked (Treatment 2.D), were presented on the material log. Stems for Treatments 2.C and 2.D were first matched for diameter, length, and curvature, and then randomly assigned to a treatment before being processed by a researcher (for details, see *St Clair et al., 2018*). The order of treatments was randomised for each subject.

Two food logs, raised on short legs to ca. 15–20 cm above ground (distance from the floor to the top of the log), were presented, containing 18 extraction holes in total: 6 small ones (ca. 9 mm wide and 70 mm deep) and 12 large ones (ca. 12 mm wide and 70 mm deep). All small holes were baited with cylindrical meat 'worms' (bored out of frozen beef heart; *St Clair et al., 2018*), and the large holes were baited with either meat worms or dead wolf spiders (Lycosidae). The allocation of the six meat worms and six spiders to the large holes was randomised across subjects but kept the same for each bird across treatments. In most trials, a teaser was positioned on each of the food logs. Trials were filmed as described above and lasted until all prey items were extracted or for 90 min, whichever occurred first.

## Video scoring and statistics

### Experiment 1

We recorded from videos what type of tool the subject (1) picked up, (2) transported to the food log, (3) deployed (i.e., inserted into the single extraction hole), and (4) used to extract the prey item with. BK scored all videos using Solomon Coder software (https://www.solomoncoder.com) in randomised order, and 15 trials (52% of the final sample, see below) were rescored by an independent observer (Matthew Steele; Cohen's kappa, for pick-up: κ = 0.53; transport: κ = 0.73; deployment: κ = 0.73; extraction: κ = 0.86). As agreement scores were lower than expected, all cases of initial disagreement were reviewed jointly by BK and MS, who confirmed BK's original scores in all instances. To ensure that all data were strictly comparable, we excluded cases from the final analyses where the tool broke before the bird had extracted the prey item with it (n = 1), or where the experimenter had made an error (n = 2): in one case, no leaf petiole was presented (Treatment 1.B), and in another, the prey had not been inserted all the way into the extraction hole and the subject managed to extract it without a tool. The final sample size was 17 birds for Treatment 1.A and 12 birds for Treatment 1.B.

To assess subjects' preferences in each treatment for behaviours (1–4) mentioned above, we conducted exact, two-sided binomial tests (random expectation for choosing the hooked stick tool: P = 1/21 = 0.0476), using the function 'binom.test' in R (*R Development Core Team, 2020*), and adjusted the p-value (Bonferroni correction) for multiple testing with the function 'p.adjust'.

## Experiment 2

From video, we recorded what type of tool the subject manufactured (Treatment 2.B) and used (all treatments) for each extraction. We did not track individual tools (so multiple tools may have been used within a trial), but confirmed that tool type matched that required by our treatments (non-hooked stick tools in Treatments 2.A and 2.C, and hooked stick tools in Treatments 2.B and 2.D). We excluded cases from analyses where this was not the case (n = 2; one each in Treatments 2.B and 2.D, where birds used a non-hooked stick tool). Following our earlier terminology (*Klump et al., 2015b*), we recorded the placement of the tool directly following each extraction (initial safekeeping) and directly before picking up the tool again after having eaten the extracted prey (final safekeeping). Since the conclusions were the same for initial and final safekeeping, only the results for final safekeeping are reported throughout. As before (*Klump et al., 2015b*), we distinguished between 'safekeeping' (tool trapped underfoot or stored in holes) and 'unsecure' placement (tool lying on the log or ground), and ran separate models (see below and Figure 3) to examine treatment effects on both the 'level' (i.e., whether or not tools were kept safe) and the 'mode' (i.e., how tools were kept safe – specifically, whether or not they were stored in holes) of safekeeping.

BK scored all videos in randomised order, and six trials (12.5%, at least one trial per treatment) were rescored by an independent observer (Mathieu Cantat; Cohen's kappa for final safekeeping: κ = 0.98); all analyses are based on the original scores. To ensure that all cases were strictly comparable, we applied the same criteria we had established for our earlier safekeeping study (*Klump et al., 2015b*) and excluded tool-placement data where the subject did not eat the prey and picked up the tool again before extracting prey from another hole (n = 39), and where extracted prey was dropped (n = 54), as this significantly influenced the level of safekeeping (generalized linear mixed model, GLMM: $\chi^2$ = 71.94, p<0.001, n = 496 tool placements) and fundamentally changed the experimental context as birds had to leave the experimental log to pick up dropped prey. The final dataset included 257 tool-placement scores from 16 birds for Treatments 2.A and 2.B combined (crow-sourced and crow-manufactured tools), and 185 tool-placement scores from 8 birds for Treatments 2.C and 2.D combined (researcher-supplied tools).

We used GLMMs ('lme4' package version 1.1-26; *Bates et al., 2015*) in R (*R Development Core Team, 2020*) with a binomial error structure and logit-link function to analyse crows' tool-related behaviours (analysed as both the level of safekeeping [i.e., whether or not tools were kept safe] and as the mode of safekeeping [i.e., how tools were kept safe – specifically, whether or not they were stored in holes]), with 'bird ID' and a combination of 'prey' and 'hole size' (three levels: meat worms in small holes, meat worms in large holes, and spiders in large holes) fitted as random effects to account for data non-independence. We first ran a model to investigate the effect of the ecologically valid combination of tool type, material, and manufacture effort on safekeeping behaviour (Treatments 2.A vs. 2.B: non-hooked stick tools crow-sourced from leaf litter vs. hooked stick tools crow-manufactured from *D. virgatus*, model #1 [level] and model #2 [mode]). Next, we ran separate models to disentangle the relative importance of tool type (Treatments 2.C vs. 2.D: non-hooked stick tools researcher-made from *D. virgatus* vs. hooked stick tools researcher-made from *D. virgatus*, model #3 [level] and model #4 [mode]; Treatments 2.B vs. 2.C: hooked stick tools crow-manufactured from *D. virgatus* vs. non-hooked stick tools researcher-made from *D. virgatus*, model #5 [level] and model #6 [mode]); tool material (Treatments 2.A vs. 2.C: non-hooked stick tools crow-sourced from leaf litter vs. non-hooked stick tools researcher-made from *D. virgatus*, model #7 [level] and model #8 [mode]); and manufacture effort (Treatments 2.B vs. 2.D: hooked stick tools crow-manufactured from *D. virgatus* vs. hooked stick tools researcher-made from *D. virgatus*, model #9 [level] and model #10 [mode]). Statistical results are reported in Figure 3, with panels for the different treatments laid out in the same way as in the schematic illustration of our experimental design in *Figure 1*, to facilitate cross-comparison.

Since Treatments 2.A and 2.B were conducted with much larger samples of crow subjects than Treatments 2.C and 2.D, we reran the latter two models with only the 8 birds that had participated in all four treatments. Since this did not change any of our conclusions, we report the results for the full datasets throughout. Model assumptions were checked using the 'testResiduals' function in the package 'DHARMa' (*Hartig, 2020*) in R (*R Development Core Team, 2020*). The significance of main effects was assessed with likelihood ratio tests at $\alpha$ = 0.05, and point estimates and 95% CIs are reported on the log-odds scale (see *Supplementary file 1c*).

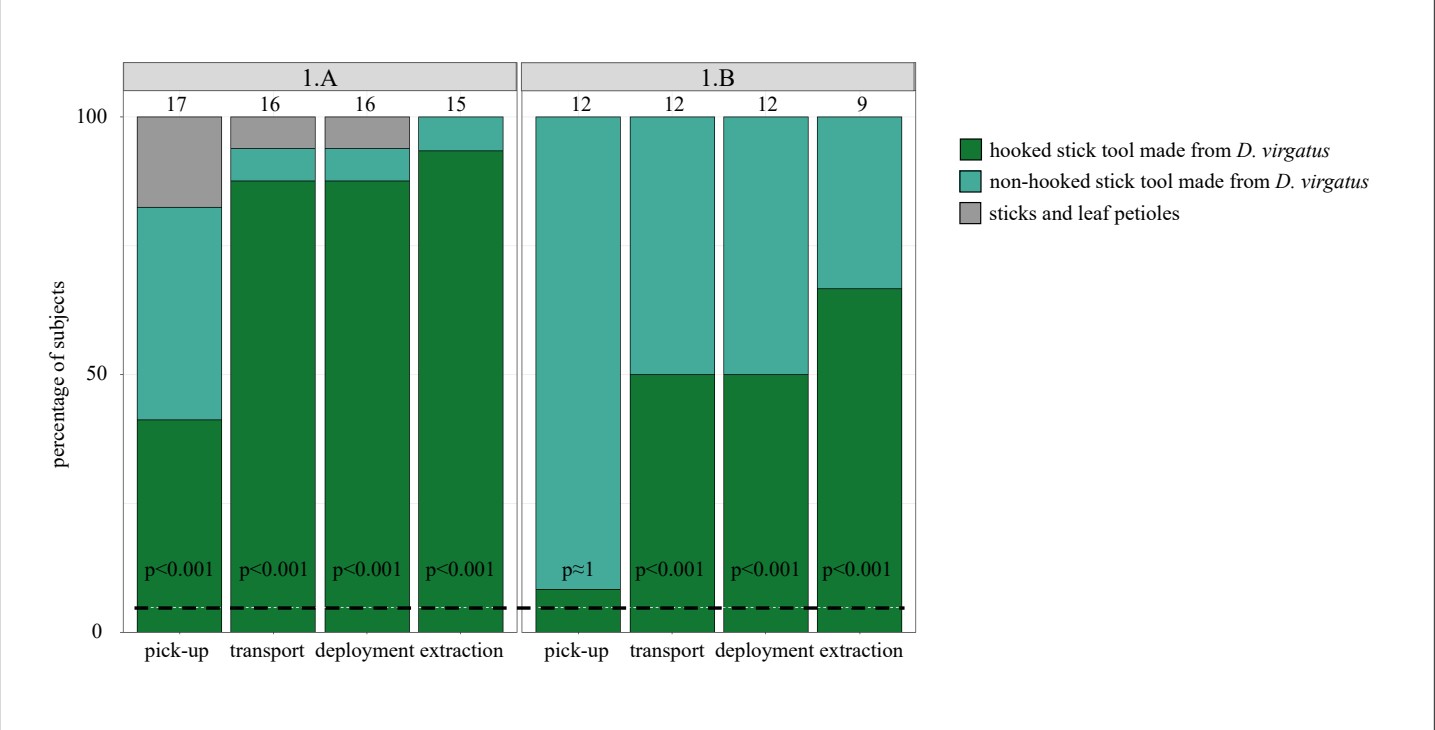

**Figure 2.** New Caledonian crows' choice of different tool types in Experiment 1. In each of the two treatments, 21 tools were presented simultaneously; Treatment 1.A: 2 stick tools researcher-made from *Desmanthus virgatus* (one hooked, one non-hooked), 19 non-hooked stick tools sourced from leaf litter (sticks and leaf petioles); Treatment 1.B: 20 stick tools researcher-made from *D. virgatus* (1 hooked, 19 non-hooked), 1 leaf petiole (for an example set of tools, see *Figure 2—figure supplement 1*). Values above bars indicate the number of crows contributing valid data; each subject contributed one datum per column. Adjusted p-values (Bonferroni correction for multiple testing) indicate the probabilities that the hooked stick tool was picked up, transported to the food log, deployed, and successfully used (extraction of bait) at the observed frequency by chance alone. The dashed line (4.76%) represents the random expectation of choosing any given one of the 21 presented tools.

The online version of this article includes the following figure supplement(s) for figure 2:

**Figure supplement 1.** Tools presented in Experiment 1 to New Caledonian crow 'APO', photographed on grid paper.

## Results

In Experiment 1, subjects showed a striking preference for hooked stick tools. In Treatment 1.A, almost all crows picked up, transported, and deployed the hooked stick tool and extracted the prey item with it (p<0.001, n = 15–17; *Figure 2*). In Treatment 1.B, crows significantly preferred the hooked stick tool for transport, deployment, and extraction (p<0.001, n = 9–12), but not at the pick-up stage (p≈1, n = 12; *Figure 2*). Interestingly, in Treatment 1.B (in which most non-hooked options were made from *D. virgatus*), fewer crows chose the hooked stick tool than in Treatment 1.A (in which most non-hooked options were leaf petioles) (*Figure 2*).

In Experiment 2, crows kept their tools safe in the vast majority of cases (*Figure 3*), both in the two naturalistic treatments (92% of 257 cases, pooled across Treatments 2.A [non-hooked stick tools crow-sourced from leaf litter] and 2.B [hooked stick tools crow-manufactured from *D. virgatus*]) and the two additional treatments with researcher-supplied tools (94% of 185 cases, pooled across Treatments 2.C [non-hooked stick tools researcher-made from *D. virgatus*] and 2.D [hooked stick tools researcher-made from *D. virgatus*]). As predicted, subjects were significantly more likely to express safekeeping behaviour (storing tools underfoot or in holes) when foraging with hooked stick tools they had manufactured from *D. virgatus* (95% of 172 cases in Treatment 2.B) than when foraging with non-hooked stick tools they had sourced from leaf litter (87% of 85 cases in Treatment 2.A; model #1: GLMM [generalized linear mixed model]: $\chi^2$ = 5.47, p=0.02, n = 257; *Figure 3*), and they also stored hooked stick tools in holes more often than non-hooked stick tools (Treatments 2.B vs. 2.A; model #2: GLMM: $\chi^2$ = 19.78, p<0.001, n = 257; *Figure 3*).

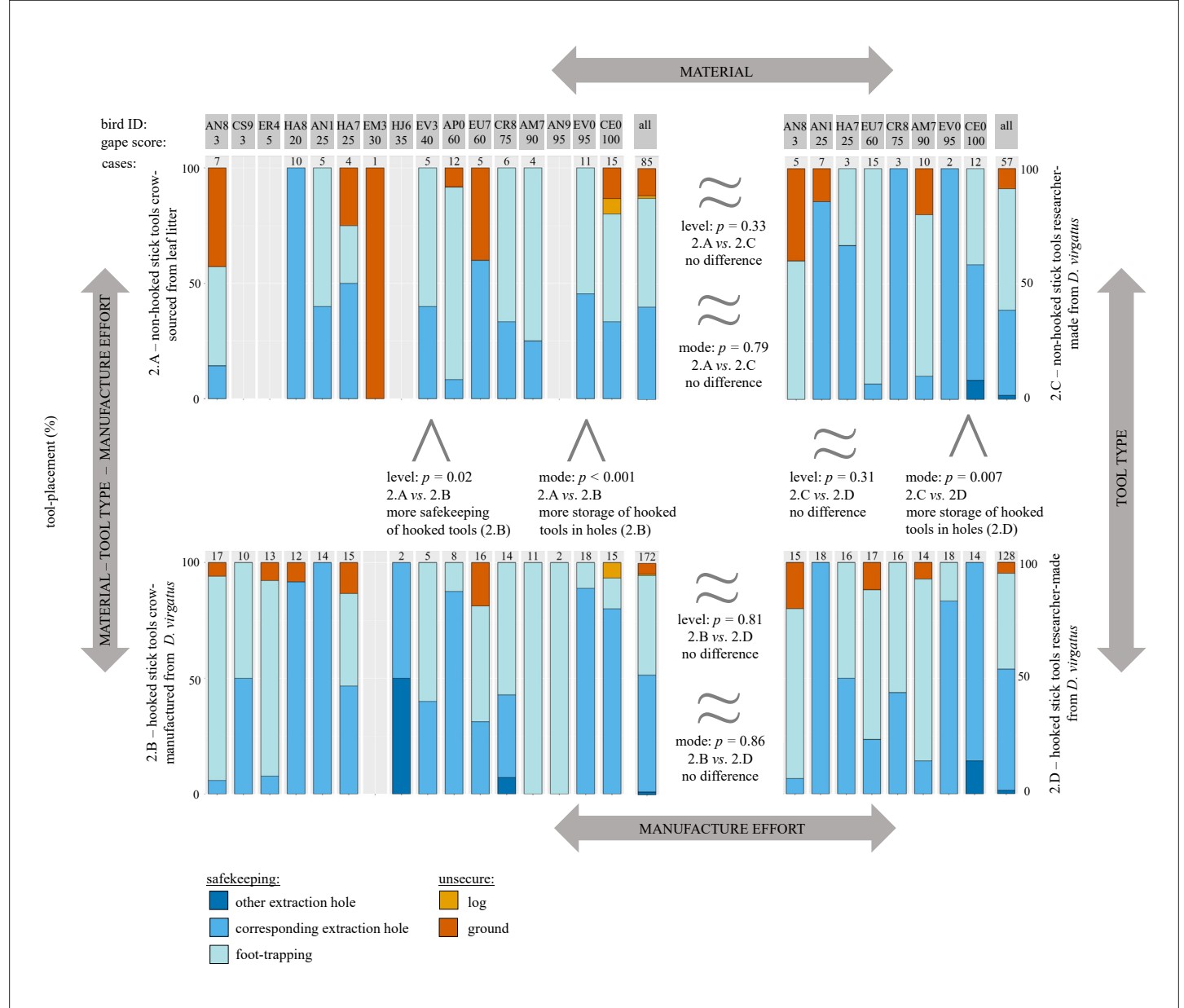

**Figure 3.** New Caledonian crows' handling of different tool types in Experiment 2. Placement of tools (percentage of cases) for crow-sourced (Treatment 2.A) and researcher-supplied (Treatment 2.C) non-hooked stick tools (shown in the top panels), and for self-manufactured (Treatment 2.B) and researcher-supplied (Treatment 2.D) hooked stick tools (shown in the bottom panels). Please see the main text and *Figure 1* (which has the same layout as this figure) for further information on treatments. Blue colours indicate secure placement of tools ('safekeeping'), while orange and red colours indicate unsecure placement. Subjects are identified at the top by their alpha-numerical ring codes, and values above bar charts indicate the number of prey extractions for a given treatment, for which tool placement was established; the rightmost bars for each treatment (marked 'all') show data pooled across all birds. Subjects are ordered by gape score (% black colouration; older birds usually have darker gapes). Results of comparisons between treatments are indicated by greater than or approximately equal signs, with p-values given for both the level (i.e., whether or not tools were kept safe) and the mode (i.e., how tools were kept safe – specifically, whether or not they were stored in holes) of safekeeping. For further details on statistical results, see main text and *Supplementary file 1c*.

The following factors by themselves did not significantly affect safekeeping behaviour: tool type (Treatments 2.C vs. 2.D; non-hooked stick tools researcher-made from *D. virgatus* vs. hooked stick tools researcher-made from *D. virgatus*; model #3: GLMM: $\chi^2 = 1.04$, p=0.31, n = 185); tool material (Treatments 2.A vs. 2.C; non-hooked stick tools crow-sourced from leaf litter vs. non-hooked stick tools researcher-made from *D. virgatus*; model #7: GLMM: $\chi^2 = 0.94$, p=0.33, n = 142); and manufacture

effort (Treatments 2.B vs. 2.D; hooked stick tools crow-manufactured from *D. virgatus* vs. hooked stick tools researcher-made from *D. virgatus*; model #9: GLMM: $\chi^2$ = 0.06, p=0.81, n = 300). For reference, the comparison between Treatments 2.B and 2.C was non-significant (model #5: GLMM: $\chi^2$ = 0.87, p=0.35, n = 229).

Interestingly, closer inspection of the data revealed that the presence of the hook alone had a measurable effect on the mode of safekeeping (i.e., how, rather than whether, tools were kept safe). Crows stored (supplied) hooked stick tools researcher-made from *D. virgatus* significantly more often in holes than (supplied) non-hooked stick tools researcher-made from *D. virgatus* (Treatments 2.D vs. 2.C; model #4: GLMM: $\chi^2$ = 7.24, p=0.007, n = 185). This increased tendency to store hooked tools in holes was also observed when comparing crows handling hooked stick tools self-manufactured from *D. virgatus* and (supplied) non-hooked stick tools researcher-made from *D. virgatus* (Treatments 2.B vs. 2.C; model #6: GLMM: $\chi^2$ = 4.11, p=0.04, n = 229). The mode of safekeeping (i.e., how tools were kept safe – specifically, whether or not they were stored in holes) was neither affected by tool material (Treatments 2.A vs. 2.C; non-hooked stick tools crow-sourced from leaf litter vs. non-hooked stick tools researcher-made from *D. virgatus*; model #8: GLMM: $\chi^2$ = 0.07, 0.79, n = 142) nor manufacture effort (Treatments 2.B vs. 2.D; hooked stick tools crow-manufactured from *D. virgatus* vs. hooked stick tools researcher-made from *D. virgatus*; model #10: GLMM: $\chi^2$ = 0.03, p=0.86, n = 300).

## Discussion

The physical properties of tools are key determinants of foraging efficiency (e.g., *Sanz et al., 2009*; *Fragaszy et al., 2010*; *Lapuente et al., 2017*; *Lamon et al., 2018*; *St Clair et al., 2018*), and many habitual tool users, including the NC crow, have been shown to exhibit some degree of tool selectivity (e.g., *Aumann, 1990*; *Chappell and Kacelnik, 2002*; *Carvalho et al., 2008*; *Visalberghi et al., 2009*; *Gumert and Malaivijitnond, 2013*; *Sirianni et al., 2015*; *Visalberghi et al., 2015*). In Experiment 1, we established that NC crows strongly prefer hooked stick tools when they have access to both hooked and non-hooked stick tools, and that they are able to differentiate between these tool types even when they are made from the same material (see also *St Clair and Rutz, 2013*). We had previously shown that, when presented with experimentally altered 'puzzle' tools (*St Clair and Rutz, 2013*), crows pay attention to the presence of a hook, prioritising it over two other, potentially efficiency-enhancing design features – a curved tool shaft and a functional end with bark removed (for discussion, see *St Clair and Rutz, 2013*; *Sugasawa et al., 2017*). While both of our choice tests clearly confirmed a strong preference for hooks, more birds chose the hooked stick tool when the non-hooked options were of a noticeably different material (Treatment 1.A), rather than when both tool types were made from *D. virgatus* and thus looked superficially similar (Treatment 1.B). This could be an indication that NC crows sometimes employ a 'rule of thumb' when selecting tools (*Hunt et al., 2006*; *Hunt, 2021*), initially paying more attention to the tool material than to the tool type (see *Klump et al., 2019* and discussion below).

Importantly, Experiment 2 demonstrated that crows' preference for hooked stick tools was reflected in their tool-handling behaviour, with hooked stick tools being kept safe more frequently than non-hooked stick tools (*Figure 3*). Tool procurement inevitably involves costs, particularly when materials are scarce or manufacture is time consuming (see 'Introduction'). To date, little is known about the magnitude of these costs, or the extent to which non-human animals – including NC crows – actually mitigate them (but see *Klump et al., 2015b*; *Auersperg et al., 2017*). The storage and re-use of tools necessarily reduces the frequency with which they must be replaced, and may allow considerable savings of time and energy. We had predicted that, given their higher procurement costs (see 'Introduction') and increased foraging efficiency (*St Clair et al., 2018*), hooked stick tools would be perceived as more 'valuable' by crows compared to non-hooked stick tools. Subjects generally looked after all of their tools very well, securing them either underfoot or inserting them into a hole – safekeeping modes we had previously documented both in captivity and in the wild (*Klump et al., 2015b*). While the overall level of safekeeping in Experiment 2 was striking (>90%), hooked stick tools were kept safe significantly more often than non-hooked stick tools. Treatments 2.A and 2.B attempted to replicate conditions crows from our study population would experience in the wild, with hooked stick tools self-manufactured from *D. virgatus* and non-hooked stick tools sourced from leaf litter. Since wild birds incur the additional cost of finding suitable plant material for tool manufacture (*D. virgatus* has a patchy distribution in our study site; unpublished data), one might expect them to

handle hooked stick tools even more cautiously than our captive subjects did; in other words, our experiment might lead us to underestimate the difference in safekeeping behaviour.

We can of course not rule out that our subjects – all confirmed hooked stick tool makers (see 'Study site and subjects') – were inexperienced non-hooked stick tool users (see *St Clair et al., 2018*). If birds do not regularly use non-hooked stick tools in the wild, they might handle this unfamiliar tool type less cautiously, which could lead to reduced levels of safekeeping during experimental trials. Although we cannot at present exclude the possibility that tool-type familiarity played a role, it is worth noting that, in Experiment 2, most birds extracted prey in each of the treatments, subjects only foraged with one tool type at any given time, and perhaps most importantly, that only scores for tool placements following successful food extractions were included in our analyses (see 'Video scoring and statistics').

In an attempt to disentangle the relative contributions of tool type, tool material, and manufacture effort, we compared treatments in Experiment 2 where tools only differed in a single aspect. When tool material and manufacture effort were held constant (i.e., all tools were researcher-supplied and made from *D. virgatus*), crows still appeared to take less care of non-hooked stick tools (91% of tools kept safe in Treatment 2.C) than of hooked stick tools (95% of tools kept safe in Treatment 2.D), but this effect was small and non-significant. This outcome may simply arise from a lack of statistical power (following *Colegrave and Ruxton, 2003*, we did not conduct *post-hoc* power analyses), or it may reflect actual crow behaviour, with subjects applying the rule of thumb of treating all tools made from *D. virgatus* as though they are hooked. While we have observed the manufacture of non-hooked stick tools from *D. virgatus* when birds are held in short-term captivity (and they appeared to treat these tools similarly to self-manufactured hooked stick tools; *Klump et al., 2015b*), non-hooked stick tools are apparently rarely made from *D. virgatus* in the wild (personal observation; *St Clair et al., 2016*). In any case, such an approach might both save time and carry little cost (*Clark and Dukas, 2003*), and would be consistent with a growing body of evidence indicating that NC crows employ simple 'heuristics' when making tool choices (*Hunt, 2021*). Crows certainly use co-occurring features as a cue for initially recognising and orienting hooked tools: as noted above, in Experiment 1, more birds chose a non-hooked stick tool when the supplied tools were stems of *D. virgatus* (Treatment 1.B) rather than different materials (Treatment 1.A), and in an earlier study, crows given puzzle tools with normally co-occurring features at different ends were more likely to pick them up in the wrong orientation (i.e., with the hooked tip pointing backwards; see *St Clair and Rutz, 2013*).

While heuristics clearly inform NC crows' initial selection of tools (*Hunt, 2021*), our experiments (present study; *St Clair and Rutz, 2013*) also demonstrate a striking ability to attend to hooks (for results for a different tool type, see *Knaebe et al., 2015*). For example, subjects that picked up a hooked stick tool in the wrong orientation will usually re-orient it very quickly before use (*St Clair and Rutz, 2013*). It thus seems unlikely that crows would assess only the material when deciding on the appropriate level of safekeeping behaviour for a given tool. In further support of this conclusion, if NC crows exclusively used material as an indicator of tool type during foraging, we would expect them to express more safekeeping behaviour when handling supplied non-hooked stick tools researcher-made from *D. virgatus* than when handling non-hooked stick tools self-sourced from leaf litter (Treatments 2.C vs. 2.A), and we did not observe a significant effect for this comparison. It is reassuring that our naturalistic (multi-aspect) comparison (Treatments 2.A vs. 2.B) remained robust when subsampled to include only those individuals that also took part in the single-aspect treatments (Treatments 2.C and 2.D), but presumably comparisons involving treatments that differed only in one aspect would have smaller effect sizes. Both 'hooked' treatments (2.B and 2.D) elicited higher levels of safekeeping than Treatment 2.C (non-hooked *D. virgatus* tools), although neither comparison was significant; while we cannot rule out at present that NC crows do not respond to these manipulations, it is conceivable that a larger sample size would have detected significant effects.

It is notable that, when the particular mode of safekeeping was examined (i.e., how tools were kept safe – specifically, whether or not they were stored in holes), rather than the level of safekeeping (i.e., whether or not tools were kept safe), tool type had a significant effect even in the absence of material differences. Specifically, researcher-supplied hooked stick tools were stored in holes significantly more often than researcher-supplied non-hooked stick tools (both researcher-made from *D. virgatus*; Treatments 2.D vs. 2.C). We have previously shown that this safekeeping mode minimises the risk of dropping a tool, and that birds store tools in holes more often when foraging at height, at ca. 1.3 m above ground, presumably to avoid retrieval costs (*Klump et al., 2015b*). While the baited log was

much closer to the ground in Experiment 2 reported here, a height of ca. 15–20 cm (from the floor to the top of the log) may have been enough to induce a change in tool-placement behaviour since birds had to leave the log to pick up a dropped tool. An interesting alternative explanation is that crows may deliberately place hooked stick tools in holes because they are then unlikely to be picked up with the hook in the non-functional orientation (see above); non-hooked stick tools have less 'functional polarity' and there is less benefit to preserving any particular orientation between spells of probing.

We had predicted that crows would show increased levels of safekeeping when foraging with self-manufactured hooked stick tools, compared to researcher-supplied hooked stick tools, as they had paid manufacture costs in the former case, but not the latter. We did not detect such an effect (Treatments 2.B vs. 2.D) and can suggest several reasons for this. The cost of manufacturing hooked stick tools within our experimental setting (Treatment 2.B) may have been too small to induce a behavioural response (it takes an NC crow only a few minutes to make a new tool, and we necessarily eliminated search costs by providing raw material; *Hunt, 1996*; *Hunt and Gray, 2004*). Moreover, these manufacture costs may have been partially compensated for by the higher value of researcher-supplied hooked stick tools in terms of both increased foraging efficiency (*St Clair et al., 2018*) and availability (only 3 tools were provided in Treatment 2.D, while 10 stems of raw material were available in Treatment 2.B). Given this arguably modest cost/benefit ratio, any resulting difference in safekeeping behaviour would also likely be small, and (as discussed above) we may have lacked the statistical power to detect it. Alternatively, there might be no difference in safekeeping behaviour for self-manufactured and researcher-supplied hooked stick tools, as crows may assess manufacture costs not on a tool-by-tool basis, but over the course of many manufactures (remembered/experienced utility; *Kahneman et al., 1997*), leading them to ascribe an equally high value to all hooked tools, rather than assessing the value of individual tools independently. At any rate, the costs of replacing a lost hooked stick tool are the same regardless of whether the lost tool was self-manufactured or serendipitously discovered; valuing a self-manufactured tool more highly than an identical 'found' tool would be logically unsound 'Concordian' thinking (where decisions are made based on past investment rather than rationally assessing the net expected future benefit; *Curio, 1987*), and perhaps we should be unsurprised that our evidence is consistent with crows avoiding this error.

While our two naturalistic treatments (Treatments 2.A and 2.B) clearly show that our subjects take better care of their hooked than their non-hooked stick tools, it remains unclear whether this difference is driven exclusively by the tool type's performance benefits (*St Clair et al., 2018*) or whether other factors – such as procurement costs – also play a role. We suggest that performance benefits are important, but cannot explain safekeeping behaviour by themselves – after all, in a 'perfect world' in which spare hooked stick tools were always within reach, there would be little motivation for a crow to keep its current tool safe. Future work should address the relative importance of these factors and how they might interact in shaping NC crows' safekeeping behaviour.

One worthwhile line of investigation may be provided by the 'larva fishing' behaviour exhibited by some NC crow populations. Since crows exclusively use non-hooked stick tools when extracting wood-boring beetle larvae from decaying candlenut logs (*Hunt, 2000*; *Bluff et al., 2010b*), this would provide a context in which non-hooked stick tools have both lower procurement costs and are functionally superior. While it would be illuminating to record safekeeping behaviour of both tool types in this foraging context in captivity, such a study would demand careful planning: if individuals that are skilled at larva fishing were unfamiliar with hooked stick tools, introducing them to this novel tool type could potentially alter the natural tool behaviour of wild populations (see *Bluff et al., 2010a*). The same ethical concerns apply to the use of artificial tools and tasks with wild-caught, temporarily captive crows. In any case, it should be possible to experimentally manipulate the procurement costs of different tool types in aviary studies by restricting access to tools or raw materials. For example, crows could be trained to expect that these resources will be withdrawn for part of the day (for work on primates, see *Mulcahy and Call, 2006*; *Dekleva et al., 2012*).

Work on the ultimate drivers of NC crows' tool safekeeping decisions should ideally go hand-in-hand with an investigation of how birds acquire the perceptions of value that arguably underpin this behaviour. Experiments should establish whether individuals' safekeeping behaviour for different tool types is dynamic, changing with the acquisition of relevant new information, while ontogenetic and social-learning studies may cast light on when, and how, safekeeping behaviour arises in the first place.

## Concluding remarks

The STRANGE framework for animal behaviour research (*Webster and Rutz, 2020*) allows us to identify several potential limitations to the generalisability of our findings. Our inferences were drawn using a sample of temporarily captive NC crows of mixed sex and age sourced from a single population using one specific trap design. The demographic composition of our test sample was fairly balanced overall (see 'Study site and subjects' and *Supplementary file 1*), and we consider it unproblematic that all our results were obtained in field aviaries as the tool safekeeping behaviour of captive and wild crows has been previously shown to be very similar (*Klump et al., 2015b*). It was important for our study that subjects were capable of manufacturing hooked stick tools, so only individuals that expressed the behaviour during pre-testing were allowed to progress to the two main experiments (see 'Study site and subjects'; of 35 potentially suitable subjects, 8 failed to interact with the task and were excluded – no adjustments were made to testing protocols to facilitate participation). As we have noted above (see 'Discussion'), exposing birds to tool types they are not naturally familiar with is ethically problematic (and was therefore avoided), but our preselection procedure inevitably has implications for the interpretation of some of our results. More generally, it is known that there is considerable variation in the tool-making behaviour of wild NC crow populations (*Hunt and Gray, 2003*; *St Clair et al., 2016*; *Steele et al., 2021*), so our specific findings may only apply to populations that manufacture hooked stick tools from *D. virgatus*. It would be fascinating to run similar experiments to assess how NC crows handle – and value – other tool types, design variants, and raw materials, and to examine if the only other known tool-using crow species, the Hawaiian crow *Corvus hawaiiensis* (*Rutz et al., 2016*; *Klump et al., 2018*), exhibits similar safekeeping behaviour.

The framing of our present experiments only assumed a relative difference in procurement costs between tool types, but made no assumptions about absolute costs. While we can be confident that it must generally be more costly for NC crows to procure hooked (*D. virgatus*) stick tools than non-hooked stick tools in our study population (see 'Introduction'), quantifying time costs is an essential next step for investigating the biological importance of tool safekeeping behaviour. Even if the additional costs of procuring a hooked stick tool are relatively small, accumulated over a bird's lifetime, they will constitute a significant investment. Importantly, these costs must be offset by enhanced tool efficiency (as demonstrated in an earlier study; *St Clair et al., 2018*) and/or by re-using tools (as explored in the present study). Charting the costs and benefits of using different tool types is key for advancing our understanding of NC crows' strikingly diverse, and regionally distinctive, tool repertoires (*Rutz and St Clair, 2012*; *St Clair et al., 2018*; *Rutz and Hunt, 2020*; see also below). We will tackle this objective in a follow-on study using extensive archive video footage our team has accumulated over the years.

While the safekeeping of tools remains patchily documented (*Nishida and Hiraiwa, 1982*; *Tebbich et al., 2012*; *Klump et al., 2015b*; *Auersperg et al., 2017*), good progress has been made with charting potential drivers of tool efficiency and tool selection behaviour in a variety of species, including birds and primates. Tool material (*Lamon et al., 2018*) and design features (*Sanz et al., 2009*; *Sugasawa et al., 2017*; *St Clair et al., 2018*) can influence foraging efficiency, and both have been shown (together with transportation distance) to affect tool selection (*Visalberghi et al., 2009*; *Sirianni et al., 2015*; *Luncz et al., 2016*). We encourage future studies that investigate whether safekeeping behaviour in these species is also sensitive to variation in tool characteristics and expected utility, as we have shown here for NC crows. It also seems worth examining the possibility that the selective storage and re-use of particularly efficient tool variants could contribute to the gradual cultural accumulation of technological innovations – an extremely rare process that has only been suggested for two non-human species, NC crows and chimpanzees (*Yamamoto et al., 2013*; *Boesch et al., 2020*; *Osiurak and Reynaud, 2020*; *Rutz and Hunt, 2020*).

There is increasing interest in how non-human animals ascribe value to resources (such as food) and objects (such as tools) (*Westergaard et al., 2004*; *Bräuer et al., 2009*; *Dufour et al., 2012*; *Evans et al., 2012*; *Auersperg et al., 2013*; *Hillemann et al., 2014*). Based on the assumption that animals assess the intrinsic quality of an item, and form preferences accordingly (*Brosnan and de Waal, 2004*), perceptions of value are often studied using an 'exchange' paradigm, where subjects are trained to either exchange a value item (e.g., food or tool) for item(s) that differ in value or amount, or learn to associate a particular reward with a non-value item (token), which can subsequently be exchanged. These studies have provided exciting insights into a variety of topics, including economic

decision-making, delayed gratification, barter, inequity aversion, and future planning in both primates (e.g., *Chalmeau and Peignot, 1998*; *Brosnan and de Waal, 2004*; *Westergaard et al., 2004*; *Bräuer et al., 2009*; *Evans et al., 2012*; *Bourjade et al., 2014*) and birds (e.g., *Dufour et al., 2012*; *Wascher et al., 2012*; *Auersperg et al., 2013*; *Hillemann et al., 2014*; *Krasheninnikova et al., 2018*), but the method usually requires extensive training of subjects, so is not normally suitable for studies with wild or temporarily captive animals (but see *Blaisdell et al., 2020*). Our new paradigm, on the other hand, does not rely on prior training and can potentially be applied not only to comparing different tool types, as we have done in the present study, but also to different variants of the same tool type, such as termite-fishing probes of different lengths, nut-cracking hammers of different sizes or weights, or tools made from different materials. Safekeeping behaviour provides a powerful proxy for the value of tools to their users, which can be studied productively with untrained subjects and allows for wide taxonomic comparisons.

## Acknowledgements

We thank the Province Sud, DENV, SEM Mwe Ara, and the late Thierry Mennesson for access to study sites and other support; Jessica van der Wal, Shoko Sugasawa, Saskia Wischnewski, Zackory Burns, and several field assistants for help with experiments and field logistics; Matthew Steele and Mathieu Cantat for rescoring some videos to assess coder reliability; Adriana Maldonado-Chaparro for statistical advice; Gustav Meibauer for valuable discussions; and Christophe Boesch, Graeme Ruxton, three reviewers (Corina Logan and two anonymous colleagues), and the editors for constructive comments on earlier drafts.

## Additional information

### Competing interests

Christian Rutz: Senior editor, eLife. The other authors declare that no competing interests exist.

### Funding

| Funder | Grant reference number | Author |
|---|---|---|
| Biotechnology and Biological Sciences Research Council | BB/G023913/1 | Christian Rutz |
| Biotechnology and Biological Sciences Research Council | BB/G023913/2 | Christian Rutz |
| Biotechnology and Biological Sciences Research Council | PhD studentship | Barbara C Klump |
| University of St Andrews | PhD studentship | Barbara C Klump |
| Biotechnology and Biological Sciences Research Council | BB/S018484/1 | Christian Rutz |
| Radcliffe Institute for Advanced Study, Harvard University | Radcliffe Fellowship | Christian Rutz |

The funders had no role in study design, data collection and interpretation, or the decision to submit the work for publication.

### Author contributions

Barbara C Klump, Conceptualization, Data curation, Formal analysis, Investigation, Methodology, Scoring of videos, Validation, Visualization, Writing – original draft, Writing – review and editing; James JH St Clair, Conceptualization, Investigation, Methodology, Visualization, Writing – original draft, Writing – review and editing, Design of the project that provided data for Experiment 2 (St

Clair et al., 2018); Christian Rutz, Conceptualization, Funding acquisition, Investigation, Methodology, Project administration, Resources, Supervision, Visualization, Writing – original draft, Writing – review and editing, Design of the project that provided data for Experiment 2 (St Clair et al., 2018)

### Author ORCIDs
Barbara C Klump http://orcid.org/0000-0003-3919-452X
James JH St Clair http://orcid.org/0000-0003-2902-4391
Christian Rutz http://orcid.org/0000-0001-5187-7417

### Ethics
All experiments, and important preparatory work, were approved by local ethical review committees at the Department of Zoology, University of Oxford, and – after the group's move – the School of Biology, University of St Andrews, and were conducted under research permits issued by local New Caledonian authorities (1341–2010/ARR/DENV, 1886–2011/ARR/DENV, 2405–2013/ARR/DENV, 2445–2014/ARR/DENV).

### Decision letter and Author response
Decision letter https://doi.org/10.7554/eLife.64829.sa1
Author response https://doi.org/10.7554/eLife.64829.sa2

## Additional files

### Supplementary files
• Transparent reporting form

• Supplementary file 1. Demographic information for New Caledonian crows trapped and tested as well as GLMM results.

• Source data 1. Source data for Experiment 2.

• Source code 1. Source code for Experiment 2.

### Data availability
Raw count data for Experiment 1 are shown in Figure 2. Raw data for Experiment 2 are shown in Figure 3 and are also supplied as a csv file.

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
