## [Editor Report]

The authors show experimentally that New Caledonian crows, a rare tool-using bird, prefer hooked stick tools for foraging and safely store them underfoot or in holes. Quantifying safekeeping behaviour helps understand how non-human, tool-using animals value different tool types.

---

## [Decision Letter]

**Decision letter after peer review:**

Thank you for submitting your article "New Caledonian crows keep 'valuable' hooked tools safer than basic non-hooked tools" for consideration by *eLife*. Your article has been reviewed by 3 peer reviewers, and the evaluation has been overseen by a Reviewing Editor and George Perry as the Senior Editor. The following individual involved in review of your submission has agreed to reveal their identity: Corina Logan (Reviewer #2).

The reviewers have discussed the reviews with one another and the Reviewing Editor has drafted this decision to help you prepare a revised submission.

We would like to draw your attention to changes in our policy on revisions we have made in response to COVID-19 (https://elifesciences.org/articles/57162). Specifically, when editors judge that a submitted work as a whole belongs in eLife but that some conclusions require a modest amount of additional new data, as they do with your paper, we are asking that the manuscript be revised to either limit claims to those supported by data in hand, or to explicitly state that the relevant conclusions require additional supporting data.

Our expectation is that the authors will eventually carry out the additional experiments and report on how they affect the relevant conclusions either in a preprint on bioRxiv or medRxiv, or if appropriate, as a Research Advance in eLife, either of which would be linked to the original paper.

Summary:

This paper experimentally shows that New Caledonian crows, a rare tool-using bird, prefer hooked stick tools and tend to keep them safe. Quantifying safekeeping behavior helps understanding how non-human, tool-using animals value different tool types.

Revisions:

All three reviewers agree that this manuscript represents a well-designed experiment that produced important results in the field. They also raise some major issues. At the subsequent consultation session, we have reached an agreement that the following three issues are particularly important and should be addressed in a revision.

(1) Text (especially in the Methods section) should be made more accessible to non-specialists of tool use (see the comments of Reviewer #1).

(2) It is unclear how costly tool manufacture is to these birds. Some empirical data are required, or the writing should be toned down if data cannot be shown (see the comments of Reviewer #2).

(3) Results should be revised by adding individual-based information (i.e., which individuals participated in which experiments). More generally, the Results section should be more explicit about what was happening to the birds (see the comments of Reviewer #3).

*Reviewer #1:*

The authors aim to investigate whether New Caledonian crows take better care of preferred tools for example by safely storing them underfoot or in nearby holes.

The experiments show a clear preference of crows for certain tool types, and the crows in a majority of trials keeping the tools 'safe' during the experimental trial.

The experiments indicate that crows attribute 'value' to tools. The authors do not test whether the observed differences in behaviour are caused by performance benefits or other factors like procurement costs.

The introduction is generally well written, however very specific about tool related behaviour in New Caledonian crow. I am wondering if somebody, not directly working on tool use and/or corvid behaviour would find this accessible. Ln 21 in the abstract mentions the idea to develop 'paradigm for productive cross-taxonomic comparisons', which I think is interesting, however I think the introduction needs to be broadened up to set the scene for this.

I find the methods section very challenging to understand. I absolutely acknowledge that presenting complex experiments with different conditions and run over several years with different samples of individuals is challenging. Maybe some sort of visual representation or flow diagram of the experimental design and procedures could help?

For example, experiment 1 is described lns. 129-136 in ln. 137 one of two conditions are introduced. Maybe this could be clarified from the beginning? Also, for a reader unfamiliar with plants of New Caledonia, the differences between the presented tools might be challenging to understand. What is for example the difference between non-hooked stick tool made from D. virgatus and non-hooked stick tools sourced from leaf litter? The authors do try to provide a rationale for experimental design, e.g. lns. 143-144, I appreciate this, but I am wondering if the experimental design could overall be clarified, so the rationale behind the presentation of different tools, etc. could be clearer?

Data availability:

I appreciate that the authors publish data and code in an accessible and appropriate format.

Statistical information provided (and more generally information about experimental design and study subjects): Appropriate.

*Reviewer #2:*

This was a well-designed study that follows on from extensive previous research on this species by investigating whether New Caledonian crows keep their preferred hooked tools safe more than their non-preferred non-hooked tools, and they do. The authors develop a paradigm for investigating how tools are valued in the wild using the validated proxy of safekeeping behavior, which will be extremely useful for bringing comparative cognition experiments to the wild in this and other species.

My main comment is that it currently does not appear to be known how costly tool manufacture is to these individuals, or more generally in other species (e.g., there are no citations for lines 40-45). It would need to be shown that making tools is costly by investigating the costs to the tool maker, which would likely be measured as their time and energy investment. Additionally, describing tool making as costly implies that there would be a high cost associated with it. The citations listed to support the statement that tool making is costly are 13, 15, and 17 (line 70). I went through these articles and found:

Citation 13: it takes a NC crow <1 min to process the hook and bend the shaft. This does not seem like it would be a high cost in terms of time investment.

Citation 15: only looked at tool preferences, not costs to the bird of making the tool.

Citation 17: deeper hooks allowed for faster prey extraction. Did not examine the costs to the bird of making a deeper hook.

Given that this article uses tool making costs as the basis for the hypotheses, it is important that it is either empirically shown that this is costly, or that the authors change the language to indicate that tool making is "likely" a costly behavior and, if so, then these hypotheses should be supported.

Ethics: please list the New Caledonian permits and their unique identifiers

lines 116 and 282-286 – when using p values, stick to the threshold of either significant or not significant (with the threshold presumably at the α=0.05 level), which means that there are no trends or biases if it is not significant (see Gibbs and Gibbs 2015 for details). P values don't show anything about effect sizes so one can't tell whether the sample size is biased toward females or whether hooked stick tools were kept safe more often overall or between treatments 2b and 2c. After correcting this, please update the discussion accordingly.

line 146 – scattered on top of what? The hooked and decoy tools? Were the hooked and decoy tools always visible even if they had other tools on top of them?

lines 213-215 – I don't understand the matching the tool type to the corresponding treatment statement. You saw which tool the bird chose and then you assigned the treatment?

lines 288 and 291 – "presence of the hook alone had a measurable effect on the mode of safekeeping" Correlational analyses were used, so causal direction cannot be inferred. I suggest changing to "association" rather than "effect".

lines 359 and 368 – add words like "likely" to acknowledge that these statements are not supported by empirical evidence (e.g., non-hooked are likely encountered only rarely; heuristics may play a role in the initial selection of tools).

line 421 – please explain the term "Concordian thinking".

line 499 – this method can work for temporarily captive individuals if there is enough time to train them (e.g., on the grackle project, we train wild-caught, temporarily captive individuals to use touchscreens and they engage in experiments on inhibition, causal cognition, and reversal learning; Blaisdell et al., 2019, Logan et al., 2019a,b). I would qualify this statement to allow this possibility by saying “so it is not suitable for studies with wild animals and can be unfeasible for temporarily-captive animals”.

discussion – regarding places where you discuss that you may have lacked the statistical power to detect differences, you could run a power analysis to find out.

References

N. M. Gibbs, S. V. Gibbs, Misuse of ‘trend’ to describe ‘almost significant’ differences in anaesthesia research, BJA: British Journal of Anaesthesia, Volume 115, Issue 3, September 2015, Pages 337-339, https://doi.org/10.1093/bja/aev149

Blaisdell AP, Seitz B, Rowney C, Folsom M, MacPherson M, Deffner D, Logan CJ. 2019. Do the more flexible individuals rely more on causal cognition? Observation versus intervention in causal inference in great-tailed grackles. (http://corinalogan.com/Preregistrations/g_causal.html) In principle acceptance by PCI Ecology of the version on 31 Jan 2019 https://github.com/corinalogan/grackles/blob/master/Files/Preregistrations/g_causal.Rmd. EcoEvoRxiv.

Logan CJ, Breen AJ, MacPherson M, Rowney C, Bergeron L, Seitz B, Blaisdell AP, Folsom M, Johnson-Ulrich Z, Sevchik A, McCune KB. 2019a. Is behavioral flexibility manipulatable and, if so, does it improve flexibility and problem solving in a new context? (http://corinalogan.com/Preregistrations/g_flexmanip.html) In principle acceptance by PCI Ecology of the version on 26 Mar 2019 https://github.com/corinalogan/grackles/blob/master/Files/Preregistrations/g_flexmanip.Rmd.

Logan CJ, McCune KB, MacPherson M, Johnson-Ulrich Z, Rowney C, Seitz B, Blaisdell AP, Deffner D, Wascher CAF. 2019b. Are the more flexible individuals also better at inhibition? (http://corinalogan.com/Preregistrations/g_inhibition.html) In principle acceptance by PCI Ecology of the version on 6 Mar 2019 https://github.com/corinalogan/grackles/blob/master/Files/Preregistrations/g_inhibition.Rmd.

*Reviewer #3:*

In this article, the authors tackle the question of safekeeping and ‘value’ New Caledonian crows may attribute to particular foraging tools through a novel experiment and the re-analysis of the results of another experiment comparing the foraging efficiency of two types of tools, some with hooks and others without.

The major strength of this manuscript lies in the general method employed, using freshly captured wild birds in a capture-then-release setting, allowing the researchers to directly test wild caught birds without attempting to their integrity. The (minor) drawback is, as acknowledged by the authors, that not all birds will be willing to participate in the experiments, and hence, will possibly affect the final sample size in the experiment. To counter this, the authors combine two experiments with the same aim, analyzing if some tools are preferred by the birds in a choice-based paradigm, and whether these tools are more likely to be safeguarded, either under the foot, or stored in one of the holes subsequently. One apparent weakness here is that it is unclear which bird participated in both experiments. For example, one might think that the 5 birds tested in 2012 participated both in Experiment 1 and 2, and the 9 2013 birds also participated in both experiments; yet there is no analysis of their behavior between the two experiments, which would have been helpful to determine whether the animals tested on safekeeping in experiment 2 expressed a very strong preference for their hooked tools in the first place in experiment 1. This does not seem a major drawback as Figure 2 overwhelmingly suggests that NC crows in this population had a strong preference for the particular material (D. virgatus), but it may be informative regarding the safeguarding of hooked tools subsequently.

There is indeed a point that remains unclear about what the birds actually value and how they actually safekeep the tools. There is first of all a clear effect of keeping one’s tool, independently of whether it is hooked or not. Even the non-hooked tools are kept 87% of time. Yet, they also appear to keep hooked tools more than non-hooked tools. Here the authors will need to be more explicit about what is really happening because the current presentation of the results does not ultimately allow one to clearly see the picture.

Overall, I think the results rather support the hypothesis of the authors that the NC crows of this population have a large preference for tools made of D. virgatus. The possibility that they overly favour hooked tools is a bit less apparent, but the authors make a good case that it may owe to the birds building a heuristic “in real life” that this particular species is going to make better tools, which are usually hooked, because of the physical constraints of the plant. The discussion on why there was no effect of manufacture was also interesting: it may indeed be that the effort required here is not massive as the shrubs are readily available to manufacture the tools.

I think this paper will be generally of interest in showing that birds, even in an experimental setting that does not favour them keeping their tools, nevertheless do so, and hence suggest that they are sensitive to the value of these tools in their everyday life. The final discussion is particularly of interest, because so far, research has mostly looked at this faculty of some animals to attribute value in ‘economical’ rather than ‘ecological’ paradigms. Once again, only a few species seem capable of attributing values to their tools, which adds ground to the importance of these findings.

– Please rephrase the results L264-300 which are currently really hard to digest. Despite re-reading several times, this reviewer just cannot get a clear picture of what is significant and what is not, and what it corresponds to in terms of safekeeping. All 4 sub-experiments are simple and make sense, but when in text we are asked to compare condition 2B to 2C, what seems obvious for the authors is much less so for the reader who has this familiarity with the paradigms, nor the time to get back to it. I also understand that safekeeping is comprised of both keeping underfoot and storing. I would suggest the authors to discuss the ‘keeping underfoot’ and ‘storing’ separately, because right now it is really hard to decipher what is what.

– I also think the authors should reorganize their presentation of ‘data trending’ in the results or “the effect was small and nonsignificant” in the discussion. As of now, it seems there are blurrying the overall message. I also got confused initially as the analysis regarding ‘treatment 2C vs 2D’ was actually several lines above than when it is evoked as trending. I also found a bit confusing tying two possible interpretations, which are not on the level: either an effect could appear with a bigger sample size, which is essentially a stat argument, or it would not because of a seemingly ecologically valid argument, that birds are acting upon a heuristic. There is ground to preregister a study with a predicted sample size to test this very hypothesis and which would allow the authors to determine whether there is really something there or not.

– Finally, I think that Figure 2 is also quite complex to follow and I wonder if the authors could find a better way to present their results. I was hoping it would help me to follow the particular result section that I have highlighted but it was not helpful to understand this (however, it could possibly help out in connecting the results between the two experiments).

---

## [Author Response]

Revisions:All three reviewers agree that this manuscript represents a well-designed experiment that produced important results in the field. They also raise some major issues. At the subsequent consultation session, we have reached an agreement that the following three issues are particularly important and should be addressed in a revision.(1) Text (especially in the Methods section) should be made more accessible to non-specialists of tool use (see the comments of Reviewer #1).

We appreciate Reviewer #1’s suggestions for making our article more accessible and provide detailed responses to their comments below. Briefly, we have: recrafted key passages in the Introduction section; spelled out all treatment names to improve readability; revised the labels of the four treatments of Experiment 2; numbered the statistical models of Experiment 2; added a supplementary figure (Figure 2 —figure supplement 1) showing an example tool set from Experiment 1; and revised Figures 1 and 3 to make them more accessible.

(2) It is unclear how costly tool manufacture is to these birds. Some empirical data are required, or the writing should be toned down if data cannot be shown (see the comments of Reviewer #2).

We are grateful to Reviewer #2 for raising this point. As noted above, measuring the precise costs of tool procurement is an important research objective, which we will tackle in a dedicated follow-on study. In the present article, we have now clarified why it is reasonable to assume – based on current evidence – that the two tool types (hooked and non-hooked stick tools) differ in procurement costs, and we have added supporting citations. We provide a more detailed response to the reviewer’s comments below.

(3) Results should be revised by adding individual-based information (i.e., which individuals participated in which experiments). More generally, the Results section should be more explicit about what was happening to the birds (see the comments of Reviewer #3).

We completely agree with Reviewer #3’s requests for additional information and have in response: added call-outs to Supplementary file 1a (which shows individual-based information) much earlier in the text than in the original version of the manuscript; and analysed whether subjects that participated in both experiments differed significantly in their preference for hooked stick tools from those that only participated in Experiment 1 (they did not). We provide a more detailed response to the reviewer’s comments below.

Reviewer #1:The authors aim to investigate whether New Caledonian crows take better care of preferred tools for example by safely storing them underfoot or in nearby holes.The experiments show a clear preference of crows for certain tool types, and the crows in a majority of trials keeping the tools ‘safe’ during the experimental trial.The experiments indicate that crows attribute ‘value’ to tools. The authors do not test whether the observed differences in behaviour are caused by performance benefits or other factors like procurement costs.The introduction is generally well written, however very specific about tool related behaviour in New Caledonian crow. I am wondering if somebody, not directly working on tool use and/or corvid behaviour would find this accessible. Ln 21 in the abstract mentions the idea to develop ‘paradigm for productive cross-taxonomic comparisons’, which I think is interesting, however I think the introduction needs to be broadened up to set the scene for this.

We agree that additional context was required here, and now clarify in the Introduction section that animals’ tool safekeeping behaviour has only recently become the focus of experimental studies. Furthermore, to highlight better the potential for innovative research on different taxa, and ultimately for cross-taxonomic comparisons, we now review anecdotal observations of safekeeping behaviour in a variety of species.

I find the methods section very challenging to understand. I absolutely acknowledge that presenting complex experiments with different conditions and run over several years with different samples of individuals is challenging. Maybe some sort of visual representation or flow diagram of the experimental design and procedures could help?

Thank you very much for pointing out that the presentation of our work needed to be improved. To help readers, we have made the layout of Figures 1 and 3 identical in terms of the four treatments used in Experiment 2, indicating clearly in both figures what the key differences between treatments are. We have also revised the labels of the four treatments of Experiment 2, to be easier to understand and compare, and numbered all statistical models. Furthermore, we have added another call-out to Supplementary file 1a, which reports individual-level participation in experiments (and different treatments) across years.

For example, experiment 1 is described lns. 129-136 in ln. 137 one of two conditions are introduced. Maybe this could be clarified from the beginning?

We have added a call-out to the two treatments.

Also, for a reader unfamiliar with plants of New Caledonia, the differences between the presented tools might be challenging to understand. What is for example the difference between non-hooked stick tool made from D. virgatus and non-hooked stick tools sourced from leaf litter?

We agree that this would be difficult to understand without first-hand knowledge of the materials or additional information. We have therefore added a supplementary figure (Figure 2 —figure supplement 1) showing the tools provided to one subject during the two treatments of Experiment 1, and added a verbal description of the main characteristics of the different tools to the figure caption.

The authors do try to provide a rationale for experimental design, e.g. lns. 143-144, I appreciate this, but I am wondering if the experimental design could overall be clarified, so the rationale behind the presentation of different tools, etc. could be clearer?

We have clarified the text passage highlighted by the reviewer and hope that adding Figure 2 —figure supplement 1 (see above) helps illustrate the rationale of our experimental design.

Reviewer #2:This was a well-designed study that follows on from extensive previous research on this species by investigating whether New Caledonian crows keep their preferred hooked tools safe more than their non-preferred non-hooked tools, and they do. The authors develop a paradigm for investigating how tools are valued in the wild using the validated proxy of safekeeping behavior, which will be extremely useful for bringing comparative cognition experiments to the wild in this and other species.My main comment is that it currently does not appear to be known how costly tool manufacture is to these individuals, or more generally in other species (e.g., there are no citations for lines 40-45).

We completely agree that the costs of tool manufacture (and use) must be considered when examining safekeeping strategies. We have recrafted key passages of the main text to address this point (see also responses below), and added citations as requested.

It would need to be shown that making tools is costly by investigating the costs to the tool maker, which would likely be measured as their time and energy investment. Additionally, describing tool making as costly implies that there would be a high cost associated with it. The citations listed to support the statement that tool making is costly are 13, 15, and 17 (line 70). I went through these articles and found:Citation 13: it takes a NC crow <1 min to process the hook and bend the shaft. This does not seem like it would be a high cost in terms of time investment.Citation 15: only looked at tool preferences, not costs to the bird of making the tool.Citation 17: deeper hooks allowed for faster prey extraction. Did not examine the costs to the bird of making a deeper hook.Given that this article uses tool making costs as the basis for the hypotheses, it is important that it is either empirically shown that this is costly, or that the authors change the language to indicate that tool making is “likely” a costly behavior and, if so, then these hypotheses should be supported.

We are most grateful to the reviewer for highlighting this issue. While tool procurement costs have not been quantified yet (see above), it is important to note that our study’s framing only assumes a relative difference in costs between tool types, but makes no assumptions about absolute costs. We can be extremely confident that it must generally be more costly for crows to procure hooked than non-hooked stick tools. In terms of sourcing suitable materials, *Desmanthus virgatus* has a patchy distribution in our study area (so crows have to search for plants or travel to known patches), while sticks are much more readily available. In terms of manufacture costs, it will take a crow more time and behavioural activity to process plant material, craft a hook, and (in some cases) add other design features (for details, see below), than to pick up or snap off a twig. Even if these additional costs are small per tool manufacture, accumulated over a crow’s lifetime they will constitute a significant investment. Importantly, these costs must be offset by enhanced tool efficiency (as demonstrated in an earlier study; St Clair *et al.,* 2018, *Nature Evol. Evol.*) and/or by re-using tools (as explored in the present study).

To clarify, we had cited papers 13, 15 and 17 in our original submission for the following reasons:

Klump *et al.,* 2015a (previously citation 13): The manufacture of a hooked stick tool involves releasing the basic tool from the stem (either by pulling or cutting), removing side branches and leaves, and often processing of the hook, stripping of bark near the functional end, and bending of the tool shaft. Although we have not yet timed these manufacturing and processing steps (see above), they all add to the overall costs of procuring a hooked stick tool.

Klump *et al.,* 2019 (previously citation 15): While this paper does not quantify costs, it highlights potential penalties for choosing the wrong plant material – in terms of increased manufacture effort, reduced tool efficiency, and/or sourcing of replacement material.

Sugasawa *et al.,* 2017 (previously citation 17): Making hooked stick tools with deep hooks tends to initially involve a two-step cut action, rather than a relatively swift, one-step pull action, so producing these more efficient variants may take extra time (see also our response below).

We now clarify these points in the brackets listing the citations and state clearly that quantifying these costs remains a challenge for the future. We completely agree that obtaining estimates of time costs is an important next step, and will produce a notable advance in our understanding of tool safekeeping behaviour (and, ultimately, technological evolution) in New Caledonian crows – as noted above, we are excited about tackling this objective in a dedicated follow-on study.

Ethics: please list the New Caledonian permits and their unique identifiers

We have now added the reference numbers of our New Caledonian research permits.

lines 116 and 282-286 – when using p values, stick to the threshold of either significant or not significant (with the threshold presumably at the α=0.05 level), which means that there are no trends or biases if it is not significant (see Gibbs and Gibbs 2015 for details). P values don't show anything about effect sizes so one can't tell whether the sample size is biased toward females or whether hooked stick tools were kept safe more often overall or between treatments 2b and 2c. After correcting this, please update the discussion accordingly.

We agree that our initial wording for reporting non-significant results was too loose and have implemented the reviewer’s suggestions. Specifically, we have: reworded the text, referring to *p*-values as either significant or non- significant (at *α* = 0.05), for the comparisons that were previously referred to as ‘trending’ (Treatments 2.C *vs.* 2.D and 2.B *vs.* 2.C); and updated the Discussion section, reporting the percentage of tools kept safe, which allows readers to see the difference between treatments (and we clearly state that this comparison was not statistically significant).

line 146 – scattered on top of what? The hooked and decoy tools? Were the hooked and decoy tools always visible even if they had other tools on top of them?

Thank you for pointing out that the description of our experimental protocol was ambiguous here. We have clarified that the remaining tools were scattered on top of the hooked and the decoy tools, while all tools remained visible*.* Please note that we have opted to call the decoy tools ‘anomalous’ tools in the revised text, which we believe is more appropriate.

lines 213-215 – I don't understand the matching the tool type to the corresponding treatment statement. You saw which tool the bird chose and then you assigned the treatment?

This is an important aspect of our methodology, which we agree required clarification. We expected crows to make/use hooked stick tools in Treatments 2.B (hooked stick tools crow-manufactured from *D. virgatus*) and Treatment 2.D (hooked stick tools researcher-made from *D. virgatus*). Since it was conceivable that crows would occasionally manufacture a non-hooked stick tool in Treatment 2.B, or snip off the hook of a supplied tool in Treatment 2.D, we carefully checked tool type for all trials at the video scoring stage. Based on this, we excluded two cases where crows had used a non-hooked stick tool. We have clarified this in the text.

lines 288 and 291 – "presence of the hook alone had a measurable effect on the mode of safekeeping" Correlational analyses were used, so causal direction cannot be inferred. I suggest changing to "association" rather than "effect".

We believe this is a misunderstanding, and respectfully maintain that we are able to establish causality in this case. This statement refers to the comparison of two experimental Treatments (2.C *vs.* 2.D) where the only difference in tool characteristics was the presence of a hook (i.e., tool material and manufacture effort were held constant). This provides experimental evidence for a causal effect of hook presence on the mode of safekeeping.

lines 359 and 368 – add words like "likely" to acknowledge that these statements are not supported by empirical evidence (e.g., non-hooked are likely encountered only rarely; heuristics may play a role in the initial selection of tools).

We apologise that it was not clear that these statements are in fact supported by empirical evidence. We have rephrased the text and added relevant references.

line 421 – please explain the term "Concordian thinking".

We have added a brief explanation.

line 499 – this method can work for temporarily captive individuals if there is enough time to train them (e.g., on the grackle project, we train wild-caught, temporarily captive individuals to use touchscreens and they engage in experiments on inhibition, causal cognition, and reversal learning; Blaisdell et al., 2019, Logan et al., 2019a,b). I would qualify this statement to allow this possibility by saying "so it is not suitable for studies with wild animals and can be unfeasible for temporarily-captive animals".

This is a fair point, and we have adjusted the text accordingly, citing Blaisdell *et al.,* (2020) as an example.

discussion – regarding places where you discuss that you may have lacked the statistical power to detect differences, you could run a power analysis to find out.

We agree that this required clarification. We did not conduct a power analysis before the experiment, because all birds that were available for testing (and engaged with set-ups) participated in our experiments, and *post-hoc* power analyses provide no more information than the *p*-value alone (Colegrave and Ruxton 2003, *Behavioral Ecology*); we now state this in the Discussion section. Following Colegrave and Ruxton’s recommendations, we have also added a table with point estimates and confidence intervals for all models (see text, and Supplementary file 1c).

Reviewer #3:In this article, the authors tackle the question of safekeeping and 'value' New Caledonian crows may attribute to particular foraging tools through a novel experiment and the re-analysis of the results of another experiment comparing the foraging efficiency of two types of tools, some with hooks and others without.The major strength of this manuscript lies in the general method employed, using freshly captured wild birds in a capture-then-release setting, allowing the researchers to directly test wild caught birds without attempting to their integrity. The (minor) drawback is, as acknowledged by the authors, that not all birds will be willing to participate in the experiments, and hence, will possibly affect the final sample size in the experiment. To counter this, the authors combine two experiments with the same aim, analyzing if some tools are preferred by the birds in a choice-based paradigm, and whether these tools are more likely to be safeguarded, either under the foot, or stored in one of the holes subsequently. One apparent weakness here is that it is unclear which bird participated in both experiments.

Thank you very much for this summary. We had provided information about subjects’ participation in the two experiments in Supplementary file 1a, but realise that this may have been difficult to find – we have now added an additional call-out to this table earlier on in the text.

For example, one might think that the 5 birds tested in 2012 participated both in Experiment 1 and 2, and the 9 2013 birds also participated in both experiments; yet there is no analysis of their behavior between the two experiments, which would have been helpful to determine whether the animals tested on safekeeping in experiment 2 expressed a very strong preference for their hooked tools in the first place in experiment 1.

We agree that it is important to ask whether the birds that were tested in both experiments differed in their preference for hooked stick tools in Experiment 1 from the birds that were only tested in Experiment 1. This matters since any such difference (e.g., birds participating in both experiments preferring hooked stick tools more than birds only participating in Experiment 1) could imply that our results from Experiment 2 are not generalisable. We have checked this, and found no significant difference (Fisher’s exact tests for pick-up, transport, deployment and extraction: all *p* ≥ 0.67); we added this to the paragraph in the Methods section where we evaluate potential sampling biases (i.e., the ‘STRANGEness’ of our test sample).

This does not seem a major drawback as Figure 2 overwhelmingly suggests that NC crows in this population had a strong preference for the particular material (D. virgatus), but it may be informative regarding the safeguarding of hooked tools subsequently.

Please see our response above.

There is indeed a point that remains unclear about what the birds actually value and how they actually safekeep the tools. There is first of all a clear effect of keeping one's tool, independently of whether it is hooked or not. Even the non-hooked tools are kept 87% of time. Yet, they also appear to keep hooked tools more than non-hooked tools. Here the authors will need to be more explicit about what is really happening because the current presentation of the results does not ultimately allow one to clearly see the picture.Overall, I think the results rather support the hypothesis of the authors that the NC crows of this population have a large preference for tools made of D. virgatus. The possibility that they overly favour hooked tools is a bit less apparent, but the authors make a good case that it may owe to the birds building a heuristic "in real life" that this particular species is going to make better tools, which are usually hooked, because of the physical constraints of the plant.

In order to explain our results better, we have: modified Figures 1 and 3 so that they are now identical in terms of the layout of the four treatments in Experiment 2; added a descriptive label on the y-axis of Figure 2; and described comparisons between treatments. We hope that these changes make it clearer that our subjects indeed favoured hooked stick tools: Experiment 1 established that crows, when given a choice, significantly prefer hooked stick tools over non-hooked stick tools, choosing the single hooked stick tool even when the vast majority (19 out of 21) of tools presented were non-hooked stick tools made from *D. virgatus*. In terms of safekeeping, crows in Experiment 2 only had access to one tool type at a time, so it is perhaps unsurprising that we see a high overall level of safekeeping behaviour. Nevertheless, subjects kept hooked stick tools safe significantly more often than non-hooked stick tools.

The discussion on why there was no effect of manufacture was also interesting: it may indeed be that the effort required here is not massive as the shrubs are readily available to manufacture the tools.

Many thanks for this comment; we thought it was important to discuss this possibility. We also agree with the reviewers and editors that it would be very interesting to quantify search and manufacture costs and are planning to do this in a follow-on study – please see our detailed response to Reviewer #2 above.

I think this paper will be generally of interest in showing that birds, even in an experimental setting that does not favour them keeping their tools, nevertheless do so, and hence suggest that they are sensitive to the value of these tools in their everyday life. The final discussion is particularly of interest, because so far, research has mostly looked at this faculty of some animals to attribute value in 'economical' rather than 'ecological' paradigms. Once again, only a few species seem capable of attributing values to their tools, which adds ground to the importance of these findings.– Please rephrase the results L264-300 which are currently really hard to digest. Despite re-reading several times, this reviewer just cannot get a clear picture of what is significant and what is not, and what it corresponds to in terms of safekeeping. All 4 sub-experiments are simple and make sense, but when in text we are asked to compare condition 2B to 2C, what seems obvious for the authors is much less so for the reader who has this familiarity with the paradigms, nor the time to get back to it.

We agree that this passage of the Results section was not clear. We have now added the revised descriptive labels for all treatments and model numbers (see above) to the text, which we hope will significantly enhance readability.

I also understand that safekeeping is comprised of both keeping underfoot and storing. I would suggest the authors to discuss the 'keeping underfoot' and 'storing' separately, because right now it is really hard to decipher what is what.

Many thanks for pointing out that this required clarification. We analysed safekeeping behaviour both in terms of whether or not a tool was kept safe (irrespective of how this was achieved) and the ‘mode’ of safekeeping (i.e., whether or not a tool was stored in a hole). We have clarified these two levels of analyses in the text, which we hope addresses the reviewer’s comment.

– I also think the authors should reorganize their presentation of 'data trending' in the results or "the effect was small and nonsignificant" in the discussion. As of now, it seems there are blurrying the overall message. I also got confused initially as the analysis regarding 'treatment 2C vs 2D' was actually several lines above than when it is evoked as trending. I also found a bit confusing tying two possible interpretations, which are not on the level: either an effect could appear with a bigger sample size, which is essentially a stat argument, or it would not because of a seemingly ecologically valid argument, that birds are acting upon a heuristic. There is ground to preregister a study with a predicted sample size to test this very hypothesis and which would allow the authors to determine whether there is really something there or not.

Reviewer #2 had noted the same point about reporting of non-significant results, and we have amended wording in the manuscript to avoid any reference to ‘trending’. We agree that running additional subjects on these treatments would be ideal, but unfortunately, we will not be able to do this for various logistical reasons. That said, we now flag more clearly in the text that the alternative explanations we offer are conceptually different (lack of statistical power *vs.* ecological reasons), as rightly noted by the reviewer.

– Finally, I think that Figure 2 is also quite complex to follow and I wonder if the authors could find a better way to present their results. I was hoping it would help me to follow the particular result section that I have highlighted but it was not helpful to understand this (however, it could possibly help out in connecting the results between the two experiments).

Thank you very much for highlighting that figure presentation needed to be improved. We assume that this comment refers to Figure 3 (rather than Figure 2) and have made the following amendments. First, we have added arrows to indicate the difference between treatments (material, tool type, manufacture effort), as we had previously done in Figure 1. This aligns Figures 1 and 3 in terms of presentation, which we hope will make it much easier for readers to eyeball the various treatment comparisons. Second, we have written the main results under each comparison, indicating the treatments being compared and the effect that was observed. Third, we have changed ‘=’ to ‘≈’ to indicate that the expressed safekeeping levels/modes (see above) in these treatments were of course not equal, but the comparisons were statistically non-significant. Fourth, we have revised the treatment labels, to be easier to understand (and these are used consistently across the main text). Fifth, we have numbered all models to facilitate comparisons across the Methods and Results sections, Figure 3 and Supplementary file 1c. And, finally, we have added a call-out to Figure 1 in the figure caption to highlight that the layout of both figures is identical.